

# Attitudes of medical students in Khartoum, Sudan towards the doctor-patient relationship: a cross-sectional study

Aya M. Haiba[1],[*] and Marwan M. Haiba[2],[*]

[1] Department of Community Medicine, Faculty of Medicine, University of Khartoum, Khartoum, Sudan
[2] Faculty of Medicine, University of Ibn Sina, Khartoum, Sudan
[*] These authors contributed equally to this work.

Corresponding author
Aya M. Haiba,
ayoyabanonat@gmail.com

## ABSTRACT

**Background/Objectives**. The doctor-patient relationship is essential to the delivery of high-quality medical care. A strong doctor-patient relationship that improves patient outcomes and satisfaction depends on effective communication. The purpose of this study was to assess medical students' attitudes toward the doctor-patient relationship during their clinical years at the University of Khartoum. We also looked at how gender and study year affected patient-centeredness.

**Participants/Setting**. The study was conducted on medical students in their clinical years from December 2020 to March 2021. Students were selected from years 3 to 6. A total of 353 medical students constituted the study sample.

**Design**. The cross-sectional study utilized the Patient Practitioner Orientation Scale (PPOS) for the measurement of student attitudes towards the doctor-patient relationship. PPOS scores are calculated as a mean score that ranges from 1 (indicating doctor or disease-centered inclinations) to 6 (indicating patient-centered or egalitarian inclinations). Medical students' demographic data was collected, including their gender, age and study year.

**Results**. A total of 313 students completed the survey (response rate: 89%). The average total PPOS score and the scores for the caring and sharing subscales for the entire cohort were $4.08 \pm 0.53$, $4.43 \pm 0.58$, and $3.72 \pm 0.72$, respectively. Female gender was significantly associated with more patient-centered attitudes ($p < 0.001$). When compared to the start of their clinical curriculum, students' attitudes were significantly more patient-centered by the conclusion of their studies ($p < 0.001$).

**Conclusion**. A satisfactory level of patient-centeredness was demonstrated by medical students at the University of Khartoum, and gender had an impact on this quality. Additional consideration should be given to the finding that students' orientations were more patient-centered in the caring dimension and less so in the sharing one. Once addressed, improvements in that area could create an environment that enhances attitudes among students in the sharing domain, with great potential gains to patients.

## INTRODUCTION

The doctor-patient relationship is critical to providing high-quality medical treatment, and effective communication is a crucial component to creating a strong doctor-patient relationship. This improves patient happiness and compliance and has a favorable impact on health outcomes (*Simpson et al., 1991*). Patient-centered communication gives the patient more power and information, as well as a role and responsibility in decision-making (*Ishikawa, Hashimoto & Kiuchi, 2013*). It is now widely acknowledged that patient-centered care is an essential element for raising the standard of healthcare delivery, patient care plans, and medical education (*Aljuaid et al., 2016*). Shorter hospital stays, fewer problems, greater patient happiness and understanding, and a decreased risk of litigation are just a few of the quantifiable advantages of effective communication associated with patient-centeredness (*Fallowfield, 2008*).

Over the years, academics and scientists have proposed several interpretations and definitions of patient-centeredness. It was advocated that a patient-centered style be compared to a disease-centered one, in which the physician uses a primarily biomedical means of providing healthcare rather than attempting to relate to and empathize with the patient (*Henbest & Stewart, 1989*; *Krupat, Yeager & Putnam, 2000*). Doctor-centered clinicians exert control over the session by selecting what is discussed, failing to provide a two-way mechanism for information exchange, and failing to allow patients to participate in decision-making. In contrast, patient-centered physicians are eager to collaborate with patients and encourage them to take an active role in their health care (*Henbest & Stewart, 1989*). To better understand the tendencies and orientations towards the various physician styles, a validated instrument known as the Patient Practitioner Orientation Scale (PPOS) has been developed to assess the extent to which people hold patient-centered attitudes. The PPOS demonstrates strong psychometric qualities and has received extensive validation against a variety of different attitudinal measures and pertinent patient outcomes, which is why we selected this instrument in our study (*Shaw, Woiszwillo & Krupat, 2012*). Created in 1999 in America, the PPOS is an 18-item instrument originally designed to be administered to either doctors or patients. The instrument measures individual attitudes toward the doctor-patient relationship along two dimensions termed sharing and caring (*Krupat et al., 1999*). Acceptable internal consistency ($\alpha = 0.73$) of the scale was reported by *Krupat et al. (2000)*. The PPOS also revealed moderate predictive validity with patient-centered measures as well as with patient satisfaction outcomes (*Shaw, Woiszwillo & Krupat, 2012*; *Krupat et al., 2000*). The instrument has been translated and validated in a number of countries (*Hurley et al., 2018*; *Ribeiro, Krupat & Amaral, 2007*; *Liu et al., 2019*; *Tsimtsiou et al., 2007*).

Despite efforts by educators to implement patient-centered curricula, there is a sizable amount of qualitative and ethnographic evidence to support the idea that the culture of medical education lays more emphasis on the biomedical mechanisms than on the matters that are relevant to patients' preferences, worries, and emotions (*Hafferty, 1998*). A growing number of doctors and medical students downplay the importance of patient-centeredness and disregard it in their daily practice. This is despite evidence suggesting

that encouraging patients to take an active role in their health care can increase the effectiveness of doctors' therapeutic activities (*Fallowfield, 2008*). To ensure the best care delivery, patient satisfaction, adherence to treatment, and perhaps improved therapeutic outcomes, patient-centeredness must be promoted as part of the training curriculum in medical schools (*Campbell & McGauley, 2005*). Incorporating patient-centeredness into medical school curricula could help future doctors provide high-quality care and create efficient health systems, but doing so requires knowledge of the levels and trends in patient-centered attitudes that exist today (*Hurley et al., 2018*). Because medical students represent future physicians, it is necessary to investigate their attitudes toward the doctor-patient relationship in order to identify the nature of the beliefs they hold (*Ahmad et al., 2015*). Various studies have come forth to describe the attitudes of medical students as they relate to patient-centeredness. Both Brazilian and American medical students were found to hold strong patient-centered beliefs (*Ribeiro, Krupat & Amaral, 2007*; *Haidet et al., 2002*). Pakistani students, however, held strong doctor-centered beliefs (*Ahmad et al., 2015*). Scholars in America and Singapore discovered that gender is one of the factors influencing patient-centeredness (*Haidet et al., 2002*; *Lee et al., 2008*). This was similar to what was discovered among medical students in China, Greece, and Sweden, but contrary to the findings in Nepal (*Liu et al., 2019*; *Tsimtsiou et al., 2007*; *Wahlqvist et al., 2010*; *Shankar et al., 2006*).

In Africa, a number of obstacles have made it difficult for patient-centered treatment to grow widely. Systems of biomedical healthcare in the region were built on the rigid, hierarchical procedures for disease control that were established by colonial powers (*De Man et al., 2016*). The current global initiatives to create vertical, disease-specific programs that prioritize easily observable outcomes further limit these systems. Nevertheless, another obvious reason why it isn't employed more commonly is the fact that healthcare workers are not trained to provide patient-centered care (*De Man et al., 2016*).

Studies of this kind in Africa have been scarce to nonexistent. It is hoped that this study will shed light on a subject that is mostly unexplored and serve as a building block for better healthcare delivery. It would also allow for the evolution of how the doctor-patient relationship is perceived. This study was carried out to better understand how medical students view the doctor-patient relationship. Its findings should help support the demand for curricula that promote patient-centeredness. The objectives of this study were to: (1) describe the attitudes of medical students in their clinical years toward patient-centered care, using the Patient Practitioner Orientation Scale (PPOS); (2) ascertain whether gender and academic year are associated with patient-centered attitudes. Previous research has implicated gender and year of study as key factors influencing patient-centered attitudes (*Krupat et al., 1999*; *Tsimtsiou et al., 2007*; *Haidet et al., 2002*).

# MATERIALS & METHODS

## Study design and participants

We conducted a descriptive, cross-sectional study at the University of Khartoum's Faculty of Medicine, where the medical program is divided into 3 pre-clinical years followed by

3 clinical years. In each of the six years, approximately 70% of the students were female. Clinical rotations involving exposure to patients begin in the second half of the third year. Exposure to patients then grows increasingly regular and frequent as students progress through their medical education. Participants were chosen from among medical students in their third, fourth, fifth, and sixth years (clinical years) of study. These students had received exposure to patients that ranged from minimal to full regular hands-on experience.

## Instrument

Data was collected using a pretested, structured, close-ended, and self-administered scale that had been previously devised and standardized. Fields designed to collect participant sociodemographic information were included. Pretesting (pilot survey) was conducted on 14 students chosen at random to test for questionnaire field understanding and practicality; their results were not part of the final sample. The questionnaire consisted of a total of 21 fields. Age, gender, and study year constituted the first three fields, respectively. An 18-item instrument that uses a 6-point Likert scale ranging from strongly agree (given a score of 1) to strongly disagree (given a score of 6), known as the Patient Practitioner Orientation Scale (PPOS), was used to measure the students' attitudes toward the doctor-patient relationship. Overall mean scores were calculated as an average of all 18 item scores and could range from 1 (doctor-centered or paternalistic) to 6 (patient-centered or egalitarian) (*Krupat et al., 1999*). All questionnaire items presented were in the English language. Along with an overall score, the PPOS gauges attitudes regarding the doctor-patient interaction on two subscales: sharing and caring. The sharing subscale consisted of questions 1, 4, 5, 8, 9, 10, 12, and 15. The caring subscale consisted of questions 2, 3, 6, 7, 11, 13, 14, 16, and 17. The responding individual's level of support for the idea that the patient and the doctor should share authority and decision-making is indicated by the sharing score. The caring score assesses how concerned a respondent is with the importance of warmth and support in the doctor-patient interaction as well as how strongly they feel the doctor should ask about psychological issues. Mean scores for each of the subscales were calculated and could, as well, range from 1 (doctor-centered or paternalistic) to 6 (patient-centered or egalitarian).

## Sampling and data collection

The survey was administered to 3rd, 4th, 5th, and 6th-year students, all of whom had had varying degrees of clinical rotation. Students in these years were especially chosen since they represent the most experienced and mature students. Their opinions would be more carefully considered than those of their first- and second-year counterparts, who have had no clinical exposure. The corresponding total number of students enrolled in each year was as follows: 350 students in their third year, 346 in their fourth year, 311 in their fifth year, and 325 in their sixth year. To ensure accurate representation of the population, probability sampling was utilized. The sample size was calculated using Slovin's formula, which amounted to 308 participants. The designated sample size of 308 was increased by an additional 15% to allow for the making up of non-responses encountered during data collection, giving a total of 353 participants. Systematic stratified random (probability) sampling technique was employed and applied to a database containing student names

obtained from the faculty administration. Stratification was based on gender and study year for the sake of adequate representation. An interval was calculated and run through the database for the selection of participants. A total of 353 students were invited to participate by filling out an online survey. The survey was sent to targeted individuals on social media platforms (WhatsApp and Telegram) due to COVID-19 restrictions on accessibility to students. The survey took an average of 5 minutes to complete. Data was collected from late December of 2020 to late January of 2021.

## Statistical analysis

Data collected was cleaned and coded in a Microsoft Excel 2019 spreadsheet and analyzed with the Statistical Package for Social Sciences (SPSS) version 23. Descriptive statistics applied included frequencies and percentages for the description of demographics and means for the description of average Likert scale responses. Assumptions of normality of distribution were assessed using Kolmogorov-Smirnov test. Probability tests were performed to examine the relationship between PPOS scores (overall PPOS, caring subscale, and sharing subscale) and demographic variables. Student's $t$-test was run to examine the relationship between gender and overall PPOS scores and that between gender and scores on the caring and sharing subscales. The difference in means across study years was compared using one-way analysis of variance (ANOVA) for overall PPOS scores and scores on the caring and sharing subscales. *Post-hoc* comparisons with the Bonferroni test were conducted to detect differences among the subgroups. P-values of 0.05 or less were considered significant.

## Ethical approval

Ethical clearance was obtained from the Ethical Committee at the Department of Community Medicine, Faculty of Medicine, University of Khartoum. Ethics approval ID: 2/2022, Com. Med. The objectives and purpose of the study were stated and explained in writing to every participant. Informed written consent was requested and obtained from all participants. The study was based on "do no harm" principles. Participants were not identified.

## RESULTS

Students from the academic years 3–6 participated in this study ($n = 353$). Of the 353 students invited to participate, 313 responded by completing the PPOS instrument. Participant response rate was 89%. Majority of the participants were female (65%). Participants' ages ranged from 20 to 27, and the mean age was $23 \pm 1.40$. Table 1 displays the distribution of students.

The average total PPOS score for the entire cohort was $4.08 \pm 0.53$. Total PPOS scores ranged from 2.39 to 5.56. Higher PPOS scores indicate more patient-centered and egalitarian attitudes towards the doctor-patient relationship. The average scores for the caring and sharing subscales for the entire cohort were $4.43 \pm 0.58$ and $3.72 \pm 0.72$, respectively. Total PPOS scores, as well as the scores for the caring and sharing subscales, differed between males and females. Female students had a higher total PPOS score (4.16

**Table 1  Demographics of sample of medical students.**

| Variable | Frequency (%) N = 313 |
|---|---|
| Age | |
| Mean | 23 ± 1.40 |
| Range | 20–27 |
| Gender | |
| Male | 110 (35%) |
| Female | 203 (65%) |
| Study year | |
| 3rd year | 61 (20%) |
| 4th year | 86 (27%) |
| 5th year | 81 (26%) |
| 6th year | 85 (27%) |

**Table 2  Overall and subscale PPOS scores by student demographic information[*].**

| Demographic variable | Overall PPOS mean ± standard deviation | Sharing subscale mean ± standard deviation | Caring subscale mean ± standard deviation |
|---|---|---|---|
| Gender | | | |
| Male | 3.93 ± 0.51 | 3.57 ± 0.68 | 4.28 ± 0.58 |
| Female | 4.16 ± 0.52 | 3.80 ± 0.72 | 4.51 ± 0.56 |
| p-value | <0.001[***] | 0.006[***] | 0.001[***] |
| Study year | | | |
| 3rd year | 3.76 ± 0.52 | 3.34 ± 0.73 | 4.18 ± 0.50 |
| 4th year | 4.09 ± 0.48 | 3.74 ± 0.65 | 4.43 ± 0.55 |
| 5th year | 4.06 ± 0.50 | 3.71 ± 0.66 | 4.41 ± 0.60 |
| 6th year | 4.31 ± 0.50 | 4.00 ± 0.70 | 4.63 ± 0.57 |
| p-value | <0.001[**] | <0.001[**] | <0.001[**] |

Notes.
[*] All scores are mean scores, $n = 313$.
[**] $P < 0.05$, one way ANOVA.
[***] $P < 0.05$, Student's $t$-test.

± 0.52) than their male counterparts (3.93 ± 0.51). Females also scored higher in the caring and sharing subscales. Upon further investigation, female gender was found to be significantly associated with the total PPOS score ($p < 0.001$), the caring subscale score ($p = 0.001$), and the sharing subscale score ($p = 0.006$). Table 2 displays these results.

Scores also differed across study years. Overall PPOS scores were lower among third-year students (3.76 ± 0.52) than among sixth-year students (4.31 ± 0.50). With the exception of a slight drop in overall PPOS score in the 5th year, overall PPOS scores showed a steady rise, and the difference in means was found to be statistically significant ($F = 14.7$, $p < 0.001$). Table 2 displays these results. Subsequent Bonferroni testing indicated higher overall PPOS scores in fourth ($p = 0.001$), fifth ($p = 0.002$), and sixth ($p < 0.001$) year students as compared with third year students. Sixth year students also demonstrated a

significantly higher overall PPOS score than fourth ($p = 0.020$) and fifth ($p = 0.008$) year students. There was no statistically significant difference in overall PPOS scores between fourth and fifth-year students ($p = 1.000$).

## DISCUSSION

To the best of our knowledge, this is the first study to be conducted in Sudan to evaluate medical students' perceptions toward the doctor-patient interaction. Comparing Sudanese medical students' scores with scores from around the world allows for a more comprehensive understanding of the attitudes displayed by Sudanese medical students. In Sudan, very little attention is directed to administering curricula that nurture and foster patient-centeredness, and the nature of medical practice is greatly impoverished in the cornerstones of ideal delivery of care. Our findings have shown that our sample of medical students exhibit patient-centered inclinations, as indicated by an overall PPOS score of 4.08. The overall PPOS score compared to those of medical students from different parts of the world is as follows: Pakistan (3.40), China (3.63), Nepal (3.7), Saudi Arabia (4.00), America (4.57), and Brazil (4.66) (*Ribeiro, Krupat & Amaral, 2007*; *Liu et al., 2019*; *Ahmad et al., 2015*; *Haidet et al., 2002*; *Shankar et al., 2006*; *Fothan, Eshaq & Bakather, 2019*). Medical students at the University of Khartoum have demonstrated patient-centeredness in every possible domain, including the overall PPOS and the sharing and caring subscales. Their scores varied from those of their Malian counterparts, which had been lower in all domains. Malian students' overall PPOS score and subscale values for sharing and caring were 3.38, 3.04, and 3.68, respectively (*Hurley et al., 2018*). Those were in contrast to the 4.08, 3.72, and 4.43 scored by our sample of Sudanese medical students in the same respective domains.

This study's findings have been consistent with what was found by researchers in America, Singapore, China, Greece, Sweden, and Brazil, where females had higher PPOS scores (*Ribeiro, Krupat & Amaral, 2007*; *Liu et al., 2019*; *Tsimtsiou et al., 2007*; *Haidet et al., 2002*; *Lee et al., 2008*; *Wahlqvist et al., 2010*). In Pakistan and Nepal, however, females were found to have the same distribution of PPOS scores as males (*Ahmad et al., 2015*; *Shankar et al., 2006*). The differences observed in this study between male and female PPOS scores (overall PPOS, caring subscale, and sharing subscale) have shown that females tend to be more patient-centered, as indicated by their higher scores in all corresponding domains. This is believed to be attributable to their better communication skills (*Roter, Hall & Aoki, 2002*).

The mean sharing subscale score (3.72) was lower than that of medical students in Nepal (3.91), Saudi Arabia (4.2), and Brazil (4.10) (*Shankar et al., 2006*; *Ribeiro, Krupat & Amaral, 2007*; *Fothan, Eshaq & Bakather, 2019*). Our students, however, outperformed medical students in China (2.88), Mali (3.04), and Pakistan (3.18), in the same respective domain (*Hurley et al., 2018*; *Liu et al., 2019*; *Ahmad et al., 2015*). The mean caring subscale score of 4.43 compares to that of medical students worldwide as follows: Nepal (3.51), Pakistan (3.63), Mali (3.68), Saudi Arabia (3.8), China (4.53), and Brazil (5.20) (*Hurley et al., 2018*; *Ribeiro, Krupat & Amaral, 2007*; *Liu et al., 2019*; *Ahmad et al., 2015*; *Shankar*
*et al., 2006*; *Fothan, Eshaq & Bakather, 2019*). Religious, cultural, and socioeconomic distinctions between countries might explain these disparities. Every country varies in its nature of expressing empathy and the extent to which emotion and feelings are relayed (*Liu et al., 2019*). For instance, in Asian cultures, patients favor physicians who are more likely to base their diagnosis and treatment plans on "family-based" or "doctor-based" considerations. This contrasts with the ethos of Western nations, where patients prefer that their doctors tell them the truth when it comes to "breaking bad news", such as how to handle end-of-life care. It is, however, comparable to the civilizations of Africa, where caring has consistently outperformed sharing in studies (*Hurley et al., 2018*; *Lee et al., 2008*; *Searight & Gafford, 2005*; *Tai & Tsai, 2003*). The ease with which doctors interact with their patients is primarily determined by cultural restrictions that govern the flow of information during the encounter. As a result, a more conservative community would have less opportunities for contact between people of various sexes, including patients and doctors, and the quality of the exchange would suffer as a consequence (*Fothan, Eshaq & Bakather, 2019*).

Our students scored higher in the caring subscale domain (4.43) than they did in the sharing subscale domain (3.72), indicating that they are more interested in caring for their patients than they are in sharing information with them. This quality has also been exhibited by students in China, where the culture there is known to prefer doctors who are more inclined to make "doctor-based" decisions on the patients' behalf, taking into consideration their psycho-social status. This is unlike the Western culture, which prefers doctors to more openly share items relating to the healthcare of patients (*Liu et al., 2019*). The finding of a higher mean caring subscale score could be explained by the possibility that students are aware of the patients' desire for empathy and the creation of connections that allow for mutual channels of understanding (*Ting et al., 2016*). However, it is widely regarded in African societies that medical personnel must interview patients with absolute authority, or their medical judgment will be called into question. As a result, decision-making is seen as being solely the responsibility of the doctor, and patient input is not necessarily valued (*Lau, Christensen & Andreasen, 2013*; *Conteh, Stevens & Wiseman, 2007*; *Moore, 2009*). Such deeply ingrained ideas can make it difficult for our medical students to better express themselves in the sharing realm.

Overall PPOS scores rose significantly with advancing school year ($p < 0.001$). This finding contradicts what was discovered among American and Greek medical students, who saw a drop in overall PPOS scores as their school year advanced (*Tsimtsiou et al., 2007*; *Haidet et al., 2002*). It was, however, consistent with findings among students in Brazil, where it was reported that students' overall PPOS scores experienced a rise across consecutive school years (*Ribeiro, Krupat & Amaral, 2007*). Scores were therefore higher among students of later years than they were among those in the earlier years. Other studies have demonstrated no change in overall PPOS scores among students across consecutive school years. Those were the studies from Pakistan, Singapore, and Sweden, which all reported that students' overall PPOS scores remained stable and did not decline throughout their years of medical education (*Ahmad et al., 2015*; *Lee et al., 2008*; *Wahlqvist et al., 2010*). This indicates that students did not become less or more patient-centered

as their years progressed. The positive association of overall PPOS score with advancing school year among our students suggests that as students advanced in their medical years, they were growing more patient-centered. It also confirms that students have not drifted away from the idealism they held in the earlier years of medical school as they became more engrossed in the biomedical aspects of disease (*Haidet et al., 2002*). Increasing levels of stress brought on by heavy workloads and responsibilities have been linked to a loss in empathy and, consequently, patient-centeredness among medical students. This burnout is a result of the psychological distress these students face (*Levinson, Lesser & Epstein, 2010*; *Damiano et al., 2016*). Yet among our pupils, this has not been the case. The rise in patient-centeredness demonstrated by our students may be attributed to their growing maturity and clinical exposure. As students delve further into clinical training and spend more time coming into contact with patients, they have come to better appreciate the value of practicing ideals that would refine their encounters with patients and boost health outcomes.

It is important to note that our study has a number of limitations. Such limitations include a restriction to one medical school. It is advised that subsequent studies should include students from a number of medical schools to allow for a broader sampling. The nature of the study design does not allow for follow-up comparisons to be made. Thus, we recommend future research consider longitudinal designs to better understand the changes in patient-centeredness experienced by medical students as they evolve in their medical undergraduate years. Prospective studies must attempt to investigate medical students' actual behaviors toward the doctor-patient relationship, as the PPOS only assesses attitudes and orientations toward that contact, not actual actions.

## CONCLUSION

Medical students at the University of Khartoum display a satisfactory level of patient-centeredness. Gender plays a role in the degree of patient-centeredness exhibited by an individual, as has also been reported in other studies. Our data also suggests that students are becoming more patient-centered as their school year advances. It was demonstrated that more work needs to be done to address the fact that students' orientations were more patient-centered in the caring facet and less so in the sharing one. This calls for further investigation into why these differences in scores exist. Once addressed, with an emphasis on building effective skills and favorable attitudes toward power sharing, reforms in this area could help establish an atmosphere to help raise the suboptimal attitudes observed among students in the sharing domain, potentially providing enormous advantages to patients.

## ACKNOWLEDGEMENTS

The authors would like to thank Dr. Edward Krupat for permitting the use of the Patient Practitioner Orientation Scale and for the very helpful material he shared with us; Dr. Elfatih Malik for his keen and constructive words and for the incredible heart and passion

he puts into teaching. The authors also thank all of the students who participated in the study for their contributions to this work.

### Funding
The authors received no funding for this work.

### Competing Interests
The authors declare there are no competing interests.

### Author Contributions
- Aya M. Haiba conceived and designed the experiments, performed the experiments, analyzed the data, prepared figures and/or tables, authored or reviewed drafts of the article, and approved the final draft.
- Marwan M. Haiba conceived and designed the experiments, performed the experiments, analyzed the data, prepared figures and/or tables, authored or reviewed drafts of the article, and approved the final draft.

### Human Ethics
The following information was supplied relating to ethical approvals (*i.e.*, approving body and any reference numbers):

Ethical clearance was obtained from the Ethical Committee at the Department of Community Medicine, Faculty of Medicine, University of Khartoum (Ethics approval ID: 2/2022, Com. Med).

### Data Availability
Datasets available at https://doi.org/10.6084/m9.figshare.21995129.

### Supplemental Information
Supplemental information for this article can be found online at http://dx.doi.org/10.7717/peerj.15434#supplemental-information.

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
