# Peer review of "Attitudes of medical students in Khartoum, Sudan towards the doctor-patient relationship: a cross-sectional study"

_PeerJ, doi:10.7717/peerj.15434_

## Round 0.1 · original submission · Major Revisions

The research question and design are not innovative; nevertheless, it may add the perspective of Sudan to the current literature. Please respond to the reviewer's comments. You are not obliged to add a figure as suggested by reviewer 1 but you need to explore the background and current literature.

There are three main aspects that you need to work on:
1- Refer to an English editor. You have short paragraphs and sometimes one sentence in a paragraph.
2- The introduction lacks a clear description of the current literature and how this study closes the gap.
3- There are some clarifications needed in the methodology and statistical analysis to better assess the validity.

I have attached a pdf with comments to help you in your endeavour.

·

Basic reporting

My entire review is contained here rather than broken down into sections:
The paper by Haiba & Haiba does not break any significant new ground, but it is an interesting contribution that complements several others using the Patient Practitioner Orientation Scale (PPOS) to study the patient centered beliefs of patients, practitioners, and medical students throughout the world. The fact that the authors collected data from medical students in the Sudan makes it a valuable contribution that extends the measurement and discussion of patient centeredness into a new culture. In addition, I appreciate that the authors make reference to how their findings are similar and different from findings in other parts of the world.
The paper is well-written overall, although I would recommend that it be edited by a fluent English speaker, as there are subtle issues of English language that need to be addressed here and there. Beyond that, I have no major critiques of the paper, even though I have a number of comments about ways in which it might be improved. These are:
• In the abstract and later in the paper, the authors describe that the students are not as “competent” in Sharing. We are in the realm of attitudes and beliefs, and therefore it seems inappropriate to refer to a low attitude score in the language of competency.
• Concerning the differences in mean score for Caring and Sharing, somewhere in their Discussion the authors might expand the finding that their students seemed open to Caring, which is to say, open to getting to know their patients and the impact of illness on them—yet were not as open to the Sharing of power and information. Is this consistent with what might expect from their knowledge of Sudanese society?
• The authors refer to 7 styles of communication, using references 2 & 3. Reference 2 is extremely old and I don’t believe that reference 3 describes styles. This should either be deleted or referenced properly.
• On lines 80-82 the authors refer to issues of congruence and drop this issue almost immediately. The congruence of doctors’ and patients’ attitudes is an interesting issue that flows from a consideration of patient-centeredness, but it is beyond the focus of this paper and should be deleted as a distraction to the main focus here.
• The study included students in years 3-6. Although this is not a flaw of the study per se, it would have been interesting to know the attitudes of students in years 1 & 2, before they are introduced to clnical medicine, especially since the authors found a tendency for students’ attitudes to change over time.
• Since the authors do note a change over time, I would like to hear an explanation offered for why they believe a change in this direction occurs, especially since the change they have detected is not consistent with some other studies in other parts of the world.
• On line 150 the authors refer to scores that are below and above 3.5 as patient-centered or not. Some people have drawn arbitrary lines as cutoffs such as 3.5, but most studies of this sort simply think of PPOS scores as continuous, not above or below a specific cutoff point. Noting that mean PPOS scores differ across cultures, this makes an arbitrary cutoff point even less valuable.
• We note on line 206 that the majority of participants were female. Are the majority of all students in years 3-6 also female, or is there a tendency for higher response rates among females? This is worth noting. Also, it would be nice if the authors could do a two-way ANOVA (sex by year in school) to see if the change in scores by year in school is similar or different for males and females. Depending on this finding, it might make for an interesting point of discussion as to how or whether male vs female students relate to patient-centeredness over time.
• At the very end the authors note strengths and limitations. It might be useful if they labeled these separately so that it’s immediately obvious which are which.

In short, I believe that this paper makes a worthwhile addition to the literature on patient-centeredness and culture. I have noted, however, several areas in which this paper ought to be improved before it is accepted for publication.

Experimental design

No comment beyond what I've said above.

Validity of the findings

No comments in addition that which I've stated above.

Reviewer 2 ·

Basic reporting

The manuscript is well written except some redundant sentences and paragraph

Experimental design

The methodology needs some improvements in the methodoloy and the analysis

Validity of the findings

Please find all my coments in the attached file

Additional comments

All mys commets are in the annotated attached file (the auhors needs to take these comments into consideration)

Annotated reviews are not available for download in order to protect the identity of reviewers who chose to remain anonymous.

Reviewer 3 ·

Basic reporting

Thank you for the submission of manuscript titled: Attitudes of medical students in Khartoum, Sudan towards the doctor-patient relationship: a cross-sectional study. In general, the methodology is similar with other previous studies, the only difference is the regionality, which caused a significant hindrance to determine study novelty. Whereas, here are my comments:

1. The introduction is generally clear and well-directed, however please discuss the application about the doctor-patient relationship type. Addition of further research is encouraged, especially to talk about which one is considered the best and what’s the benefit/drawback of employing these specific relationship
2. Also discuss the seven stars doctor principles and its association with doctor-patient relationship
3. Discussion on line 97-105 can be integrated to line 89-91, but make it briefer, talk about the discrepancy between the hope and available curriculum, while discussing about the real benefit of forgotten aspects in the medical education process
4. Last and third last paragraph of intro can be merged, and I don’t think that the aim must be comprehensively
5. State clearly about which region (…Studies of the kind in the region have proven scanty to nonexistent)  Sub-Saharan/Africa/Sudan itself
6. I recommend to make a chart regarding doctor patient relationship (example: https://journals.plos.org/plosone/article/figure?id=10.1371/journal.pone.0077579.g001), it is not exact like that, but for X and Y axis the authors can describe about the main trait regarding two-way communication (e.g. friendship or as applicable) and one-way communication (e.g. commander or as applicable), make it in the methods.

Experimental design

1. Instead of exact amount, just state the range or remove the number of student. Also, “6 years” was written twice, which is redundant
2. Just discuss about rotation related with doctor-patient relationship (and also it must be made briefer than the current version
3. In the questionnaire, the value range from 1-6, without 0.5 difference between the number, please recheck the statement: Scores higher than 3.5, instead of more than 3 (or it is applicable for overall measurement, not individual variable)?
4. Likert, not Lickert
5. Mention supplementary files, such as for describing question
6. The systematic random sampling must be explained in detail, such as where is the data pool of all students stored? Procedure to maintain similar distribution of students between each group, etc. An alternative sampling method, multistage random sampling maybe suitable for this study.
7. State about normality of data and provide it before deciding to use specific analytic test

Validity of the findings

1. Rewrite table one, it is suggested not to make a cross-tabulation, add also the average age in it (with standard deviation)
2. Do the post-hoc test for PPOS score based on year of study and discuss it
3. The term “association” is incorrect, ANOVA is a comparison test, to do that, I also suggest to do the Chi-square test analysis to determine association of the variable, but the cut-off to determine “patient or physician centered must be determined thoroughly” (researchers can also use the previously determined data).
4. Change p=0.000 to <0.001
5. Age and PPOS both are numerical variable (as inferred in the description), thus it is based on correlation, not association. In addition, please provide the evidence of this statement and insert it in the table
6. I suggest to make a table to make the comparison about PPOS score between country to make a better comprehension of the results
7. Discussion: This finding of higher mean Caring subscale score could be explained by the fact that students might be aware of the importance of empathy… (it is not explained based on the study findings, in general it is vague about its relation with study variables or previous study). Also, no citation on it.
8. Conclusion: Patient centeredness was found to be positively associated with overall PPOS scores (I think that is not the conclusion of this study as it is already mentioned long time ago), so remove it

Additional comments

1. There are several misconception of analysis terms, so it must be corrected with appropriate statistical analysis
2. Incorporate strength and limitation in the discussion, not separated (be aware of redundancies as the authors have discussed about cross-sectional nature of the study). Then, discuss it independently, since it could cause confusion to the reader.
3. Add long form first before the abbreviation, such as KSA, US, etc.
4. Please recheck the manuscript thoroughly, such as researches (very uncommon), several redundancies, grammatical problem (It was also shown that more work needs to be done…  The lower score in the sharing subscale needs to be addressed and
warrants further investigation to determine the cause of this difference)
5. No citation in the conclusion, remove citation number 17
6. Hidden curriculum is too speculative and have no direct association with the manuscript

---

## Round 0.2 · Major Revisions

Dear authors

Thank you for addressing our previous comments. However, it still required further modifications to make sure the findings are valid. Please attend to the English formatting and grammar and structure of the sentences. I urge you to send it to an editor. You need to attend to some comments regarding the methods. I have put detailed comments in the attached manuscript.

Reviewer 3 ·

Basic reporting

I have no other comments. Thank you for addressing my comments.

Experimental design

No comment

Validity of the findings

No comment

Additional comments

No comment, it can be accepted

---

## Round 0.3 · Minor Revisions

Dear authors,

Thank you for addressing all the comments in a professional way. I still have three minor comments.

1- I suggest changing the word "decent" in the abstract conclusion to" satisfactory"

2- table 2 is not part of your results. It should not be included. It is a summary of the literature.

3- it does not seem that the sample is stratified by gender as you don't have equal numbers. Please clarify.

4- I suggest naming the social media platforms, WhatsApp and Telegram.

5 - p- values in table 3 do not follow PeerJ guidelines. For example p=0.000. Moreover, p-values for the association between the score and gender is different between the abstract and the results section.

---

## Round 0.4 · accepted · Accept

I would like to thank the authors for their patience. The authors have addressed all the editor's and reviewers' comments successfully. The manuscript is ready for publication.

In Table 1, there is a missing decimal point in the mean of age. 23 ± 1.4